# Training Flexible Models of Genetic Variant Effects from Functional Annotations using Accelerated Linear Algebra

**Alan N. Amin** [* 1]   **Andres Potapczynski** [* 1]   **Andrew Gordon Wilson** [1]

## Abstract

To understand how genetic variants in human genomes manifest in phenotypes – traits like height or diseases like asthma – geneticists have sequenced and measured hundreds of thousands of individuals. Geneticists use this data to build models that predict how a genetic variant impacts phenotype given genomic features of the variant, like DNA accessibility or the presence of nearby DNA-bound proteins. As more data and features become available, one might expect predictive models to improve. Unfortunately, training these models is bottlenecked by the need to solve expensive linear algebra problems because variants in the genome are correlated with nearby variants, requiring inversion of large matrices. Previous methods have therefore been restricted to fitting small models, and fitting simplified summary statistics, rather than the full likelihood of the statistical model. In this paper, we leverage modern fast linear algebra techniques to develop DeepWAS (Deep genome Wide Association Studies), a method to train large and flexible neural network predictive models to optimize likelihood. Surprisingly, we find that larger models only improve performance when using our full likelihood approach; when trained by fitting traditional summary statistics, larger models perform no better than small ones. We find larger models trained on more features make better predictions, potentially improving disease predictions and therapeutic target identification.

## 1. Introduction

To predict the risk of genetic disease and understand its molecular causes, Genome Wide Association Studies (GWAS) use data from up to hundreds of thousands of individuals to build models that correlate the presence of genetic variants with phenotypes such as disease or height (Yang et al., 2010; Visscher et al., 2017; Halldorsson et al., 2021). However there are orders of magnitude more variants than measurements, making GWAS underpowered to predict phenotype or determine the effects of all but the most impactful variants.

To increase prediction accuracy and uncover the molecular causes of disease, geneticists have leveraged the fact that complex phenotypes are extremely polygenic – that is, they are affected by a huge number of variants spread throughout the genome (Manolio et al., 2009; Boyle et al., 2017). Geneticists look for features that distinguish variants that do and do not effect a phenotype on a set of chromosomes and use these features to build "functionally informed" priors to analyze variants on other chromosomes (Gusev et al., 2014; Finucane et al., 2015; Kichaev et al., 2019). To build these priors they use functional genomic features (ENCODE Project Consortium, 2012; Lizio et al., 2015), such as measurements of DNA "accessibility" or binding of transcription factor proteins near the variant; and comparative genomics features (Cooper et al., 2005; Pollard et al., 2010), such as whether the variant is in a region of the genome that is conserved across primates. As more accurate measurements of genomic features are made and more individuals have their genomes sequenced, in principle, geneticists should be able to build more accurate functionally informed priors with more flexible models that learn from more features.

In practice, however, significant computational challenges have prevented the development of large models. Functionally informed priors are typically phrased as priors on the effect of each variant in a hierarchical Bayesian model of the genetic and phenotypic data (Loh et al., 2015; Zheng et al., 2024). Ideally, we could fit the prior using an empirical Bayes approach to maximize the marginal likelihood (Ni et al., 2018). Unfortunately this is numerically challenging due to linkage disequilibrium (LD) – the presence of variants in the genome can be strongly correlated, and accounting for this correlation in the marginal likelihood involves inverting and calculating the log determinant of a large matrix known

---

*Equal contribution [1]New York University. Correspondence to: Alan Amin <alanamin@nyu.edu>.

*Proceedings of the 42nd International Conference on Machine Learning*, Vancouver, Canada. PMLR 267, 2025.

as the LD matrix. To avoid inverting this matrix, state-of-the-art methods 1) fit simple parametric models of the relation between functional annotations and phenotype, and 2) sacrifice statistical efficiency by fitting summary statistics or an approximation of the marginal likelihood (Finucane et al., 2015; Li et al., 2024).

A similar challenge of having to invert a large matrix to perform empirical Bayesian inference was addressed in works on Gaussian process regression with two strategies (Gardner et al., 2018). First, using an iterative algorithm, inversion of an $M \times M$ matrix could be reduced from $\mathcal{O}(M^3)$ to $\mathcal{O}(M^2K)$ where $K << M$ is the number of iterations. Second, by rearranging the problem so that the large matrix is well-conditioned the number of steps $K$ could be reduced by orders of magnitude.

Here we introduce a method to train large models that predict variant effects from functional annotations – Deep genome Wide Association Studies (DeepWAS) (Fig. 1). We outline our contributions:

- We propose a **method to train large neural networks** on phenotype association data with millions of variants by leveraging a banded approximation to the LD matrix and using the approximating slices as mini-batches.

- We train **models that maximize likelihood rather than fit summary statistics or approximations.** To do so, we rearrange our likelihood to make it amenable to acceleration from iterative linear algebra algorithms, allowing us to efficiently perform challenging linear algebra operations at each training step.

- We **curate a large set of genomic features** to train our models.

- We train large functionally informed priors on large public phenotype association data. We see that, when fitting summary statistics (LD score regression), larger models fit the data worse than small models. However, **when maximizing likelihood with DeepWAS, larger models fit the data better than small models.**

- We show that **larger models trained on more features achieve better data fit**, suggesting that even larger models may yield further improvements.

Our code for training DeepWAS models is available at https://github.com/AlanNawzadAmin/DeepWAS.

## 2. Background

In this section we explain how to train models that describe phenotypic traits using variants in the genome as features.

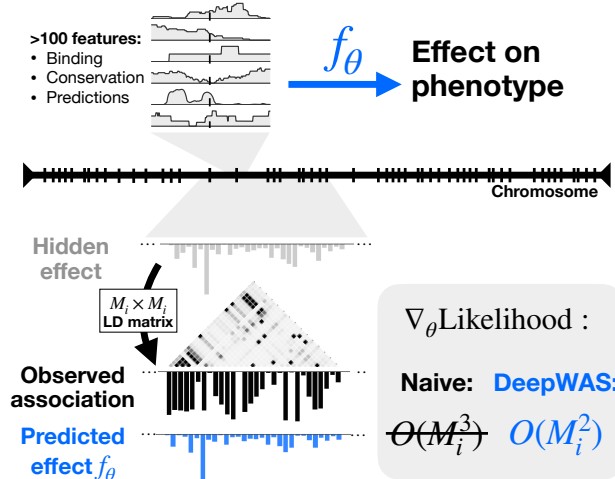

Figure 1. **DeepWAS enables training large models to predict the effect of variants from genomic features by leveraging fast linear algebra.** Top: We want to train a model, $f_\theta$, to predict the effect of a variant in our genome from a large set of curated genomic features in a window around the variant. Bottom: We train $f_\theta$ to maximize the likelihood of observed associations between variants and traits. We efficiently compute the likelihood by applying accelerated linear algebra on the correlation matrix of variants in a sliding window. See section 4 for full details.

In Sec. 2.1, we introduce a hierarchical Bayesian model of heritability with a functionally informed prior on the effects of individual variants. We describe how to fit such a model using public data. Then, in Sec. 2.2 we describe LD score regression, the state-of-the-art method to fit such a model in practice.

### 2.1. Functionally informed priors to predict variant effect

To learn the heritibility of a trait, suppose that we have measured the genotypes of $N$ ($\approx 10^5$) subjects – we have measured the presence or absence of variants at $M$ ($\approx 10^6 - 10^8$) positions on both chromosomes – to get a genotype matrix $\tilde{X} \in \{0, 1, 2\}^{M \times N}$, and the presence of the trait to get a phenotype vector $y \in \mathbb{R}^N$. We can assume $y$ is centered to have mean 0 and variance 1 and $X$ is a centered $\tilde{X}$ with all rows mean 0 and variance 1.

Measured traits that we are interested in, such as height, smoking status or schizophrenia, are polygenic – they are influenced by many variants scattered throughout the genome rather than a small number of positions (Manolio et al., 2009; Boyle et al., 2017). This is captured by the infinitesimal model in which each variant has a small effect drawn from a prior (Barton et al., 2016; Trippe et al., 2021).

A popular infinitesimal model is the linear model[1] $y = X^\mathsf{T}\beta + \epsilon$ with iid noise $\epsilon \sim \mathcal{N}(0, \sigma^2 I)$ where the effect size at position $m$, $\beta_m$, is independently drawn from a prior normal distribution $\beta_m \sim \mathcal{N}(0, f_m)$. Therefore we can describe the marginal distribution of $y$ as

$$y \sim \mathcal{N}\left(0, X^\mathsf{T}FX + \sigma^2 I\right), \text{ where } F = \mathrm{diag}(f). \quad (1)$$

Our first goal is estimating the effect size $\beta$. The challenge is that there are many more variables than observations, $N << M$, so it is challenging to get enough statistical power to predict the values of many $\beta$. Our second goal is to identify the features that characterize variants $m$ with large effect sizes $\beta_m$.

We can make progress towards these goals with a good prior $f$, which will increase our power to determine $\beta_m$ and predict which variants are expected to have large magnitude since $\mathbb{E}\beta_m^2 = f_m$. To build such a prior, we can take advantage of large datasets of genomic features $C_m$ (elaborated in Sec. 5), to predict $f_m$ with a model with parameters $\theta$, $f_\theta(C_m)$. Naively, we may train $f_\theta$ and $\sigma^2$ by maximizing the marginal likelihood of Eqn. 1.

**Public Statistics**    However, to protect the privacy of study participants, we are not given the precise value of $y$ and $X$; rather we are given another set of public statistics. In particular, we are given the empirical correlation matrix known as the "Linkage Disequilibrium (LD) matrix" $R = \frac{1}{N}XX^\mathsf{T}$; and the empirical associations $\hat{\beta} = \frac{1}{N}Xy$. We can then write Eqn 1 in terms of public statistics with $\sigma_N^2 \equiv \frac{1}{N}\sigma^2$:

$$\hat{\beta} \sim \mathcal{N}\left(0, RFR + \sigma_N^2 R\right). \quad (2)$$

The second term in the variance, $\sigma_N^2 R$, comes from spurious correlations with the noise $\epsilon$; if the presence of two variants $m$ and $m'$ are correlated ($R_{m,m'}$ is large) then the associations $\hat{\beta}_m$ and $\hat{\beta}_{m'}$ will have similar correlations with the noise $\epsilon$. The first term in the variance comes from the effect variants have on the trait. Specifically, the $m, m'$ entry of $RFR$ is $\sum_k R_{m,k}R_{m',k}f_k$, which is large if there are variants $k$ correlated to both $m$ and $m'$ – large $R_{m,k}$ and $R_{m',k}$ – which are expected to have large effect $f_k$.

Now, in principle, we could build a prior by maximizing the likelihood of Eqn. 2 using gradient descent:

$$-\frac{1}{2}\hat{\beta}^\mathsf{T}\left(RF_\theta R + \sigma_N^2 R\right)^{-1}\hat{\beta} - \frac{1}{2}\log\left|RF_\theta R + \sigma_N^2 R\right| + c \quad (3)$$

where $c$ is a constant value. The challenge is the need to 1) calculate all $M$ non-zero entries in $F_\theta$, which becomes challenging for large models, and 2) invert and calculate the log determinant of the huge $M \times M$ matrix.

### 2.2. LD score regression (LDSR)

One way to solve problem 2 is to avoid the need to invert a large matrix by only fitting summary statistics. From Eqn 2 we can note that a variant $m$ expected to have a large association if it is correlated to other variants expected to have large effects[2]:

$$\mathbb{E}[N\hat{\beta}_m^2] = N\sum_{m'} f_{m'}R_{mm'}^2 + \sigma^2. \quad (4)$$

The simplest model of heritability gives each variant the same expected heritability, $f_m = f$, in which case we can write Eqn. 4 as $\mathbb{E}[N\hat{\beta}_m^2] = Nf\left[\sum_{m'} R_{mm'}^2\right] + \sigma^2$. The term in the brackets, known as the LD score, measures how many other variants $m$ is correlated with and can be precomputed before fitting $f$.

By fitting a line to the magnitudes of the association statistics $N\hat{\beta}_m^2$ and precomputed LD scores, one can recover $Nf$ as the slope and $\sigma^2$ as the intercept. This method, known as LD score regression, gives accurate predictions of how much of a trait is explained by genetics, $f$, and how much is caused by noise or the environment, $\sigma^2$ (Bulik-Sullivan et al., 2015). Finucane et al. (2015) extended this approach to fit a linear $f_m$ that depends on $d$ genomic features – in this case one performs a multi-dimensional linear regression with $d$ precomputed variables. Unfortunately, LD score regression loses statistical efficiency by not making use of correlations between $\hat{\beta}$ (Ni et al., 2018).

## 3. Previous work

**Training functionally informed priors**    Training a large neural network as a functionally informed prior by directly optimizing the likelihood of the data has, up until now, been computationally prohibitive due to the cost of linear algebra operations on the LD matrix. Previous methods have used a number of strategies to restrict the flexibility of their prior or looked at other approximate or derived objectives in order to do inference. First, most GWAS methods pick their prior with only a handful of parameters (usually 1 or 2) and fit it by grid search or other bespoke methods that struggle to scale (Yang et al., 2010; Loh et al., 2015; Speed et al., 2017; Spence et al., 2022). Second, Finucane et al. (2015) fit a linear prior by performing LD score regression in Eqn. 4. Third, Lu et al. (2016) and Fabiha et al. (2024) fit a small model by teaching it to classify the small number ($\approx 2000$) of available high confidence positive and negative causal variants. Fourth, Li et al. (2024) considered fitting a simple generalized linear model $f_\theta$ by approximating Eqn. 3 using an approximation of $R^{-1}$. All of these methods run on CPU and use parallelism to compute the gradient of the likelihood

---

[1]We ignore fixed effects such as age, sex, and population stratification in this model, assuming they have been projected out of $y$ and $X$ (Loh et al., 2018).

[2]LD score regression can also be derived in infinitesimal models more general than Bayesian linear models with a normal prior (Bulik-Sullivan et al., 2015).

across the entire genome for each update.

Unfortunately these methods are unsuitable for training a large flexible prior as they 1) require prior values for all variants in the genome for a single gradient update, making it challenging to train a large model or 2) lose statistical power by fitting summary statistics or approximations to the likelihood. In contrast, our method DeepWAS 1) updates the model using its predictions in minibatches, and accelerates linear algebra operations in each mini-batch with GPUs, 2) directly optimizes the likelihood of data from millions of variants.

In related work, Huang et al. (2024) fit a graph neural network of variants to predict $\hat{\beta}$ directly; they use their model to increase power to find more associated variants. However such a model does not distinguish between variants with large effects $\beta$ and variants they are associated with.

**Downstream uses of functionally informed priors** A number of works have built methods to use functionally informed priors to increase the power of downstream analyses. Huang et al. (2024) and Kichaev et al. (2019) demonstrated that models that can predict the effect of variants can improve the power of GWAS. Weissbrod et al. (2020) demonstrated such models can also identify causal variants and Li et al. (2020) used such variants to identify causal genes. The DeepWAS prior can in principle fit into these same pipelines.

**Flexible models of heritability** In addition to more flexible models predicting variant effects from functional annotations, we can improve fits to association data with models that are more flexible than mixed linear models. Zhang et al. (2021) consider different, non-normal, priors, and Loh et al. (2015) consider mixture of normal priors on the effect sizes. There have also been a number of nonlinear models for predicting $y$ from $X$ (Conard et al., 2023). For simplicity, DeepWAS considers the popular normal prior with a linear model and leaves more flexible models to future work.

**Fast linear algebra with large genotype matrices** Large linear algebra problems appear throughout population genetics. As such, a number of other works have looked at approximately inverting large matrices in genetics. Loh et al. (2015) used a conjugate gradient algorithm to invert the matrix of correlations of variants between study individuals – the empirical kinship matrix $XX^\intercal$; Loh et al. (2018) noted that their algorithm converges much faster after removing the top eigenvalues of the kinship matrix, improving its condition number. Berisa & Pickrell (2016) approximated $R$ with a block diagonal matrix, Shi et al. (2016) approximated $R$ with a low rank matrix, and Salehi Nowbandegani et al. (2023) approximated the inverse of the $R$ with an extremely sparse matrix; these works use these approximations in

place of the true $R$. DeepWAS uses an iterative algorithm to perform linear algebra operations on the exact matrix; we rearrange our loss so it is amenable to acceleration from iterative algorithms.

**Fast linear algebra for fitting large Bayesian models** Fitting Gaussian processes similarly involves inverting a large matrix known as the Gram matrix. While one can avoid inverting the matrix with variational inference, state of the art methods invert the Gram matrix with an iterative algorithm with a Nyström preconditioner (Gardner et al., 2018). We build a bespoke preconditioner leveraging the structure of LD matrices to quickly invert LD matrices with iterative algorithms; in our setting, our preconditioner performs much better than a general purpose Nyström preconditioner (Frangella et al., 2021).

## 4. Efficient training of the likelihood

Our goal of directly optimizing the likelihood in Eqn 3 comes with two challenges. First, in contrast to previous methods, we train a neural network model for $f_\theta$ with millions of parameters, so computing $f_{\theta,m}$ for every variant $m$ is prohibitively expensive. Second, we must perform expensive linear algebra operations like inverting and calculating the log determinant of $A_\theta = RF_\theta R + \sigma_N^2 R$, at every step[3].

First in Sec. 4.1, we utilize a banded approximation of $R$ which allows us to amortize the training of $\theta$ by treating each slice as a mini-batch. Then in Sec. 4.2 we reduce expensive linear algebra operations on $A_\theta$ to operations on a well-conditioned matrix, which can quickly be performed using iterative algorithms.

### 4.1. Using submatrices for mini-batching

A key challenge in calculating Eqn 3 is computing $f_{\theta,m}$ for every $m$ in the genome. To address this, we first break the genome up into 2700 windows of size one million and assume the associations $\hat{\beta}$ in each window are generated independently. This can be justified by the fact that $R$ is approximately block diagonal, so there is little correlation between $\beta$ further than one million positions apart (Berisa & Pickrell, 2016; Salehi Nowbandegani et al., 2023). Eqn 3 then becomes

$$\sum_i \hat{\beta}_{(i)}^\intercal (A_\theta^{(i)})^{-1} \hat{\beta}_{(i)} + \log|A_\theta^{(i)}| \qquad (5)$$

Next, note that $A_\theta^{(i)}$ takes the following form

$$A_\theta^{(i)} = R_{(i),:} F_\theta R_{:,(i)} + \sigma_N^2 R_{(i),(i)}$$

___
[3]Here we focus on fitting $\theta$ ignoring $\sigma$. Typically $\sigma$ can be estimated accurately using other simpler methods (Bulik-Sullivan et al., 2015)

where $R_{(i),:}$ represents the rectangular submatrix of $R$ whose rows are variants in window $i$ and $R_{(i),(i)}$ is similar. Calculating $A_\theta^{(i)}$ still requires calculating $f_{\theta,m}$ for every variant $m$. To reduce this calculation, we then use the well-established fact that variants that are distant in the genome should have little correlation, and so we can use a banded approximation of $R$ (Bulik-Sullivan et al., 2015); in particular, we assume that $R_{k,r} = 0$ when the positions of the $k$-th and $r$-th variants are more than one million apart. Thus,

$$A_\theta^{(i)} \approx R_{(i),(i)^+} F_\theta^{(i)} R_{(i)^+,(i)} + \sigma_N^2 R_{(i),(i)}$$

where $(i)^+$ is the set of all variants within $10^6$ positions of a variant in window $(i)$ and $F_\theta^{(i)}$ is the $(i)^+ \times (i)^+$ submatrix of $F_\theta$. Eqn 5 then allows us to optimize $\theta$, through stochastic gradient descent, by sampling windows $(i)$ and only calculating $f_{\theta,m}$ for the roughly $10^4$ variants in $(i)^+$.

**Connection to LD score regression**    Due to the large size of our windows, we expect both approximations above to be accurate. In contrast, if we focus on the extreme case of a window size of 1 the objective, Eqn. 5 becomes

$$\sum_i \frac{N\hat{\beta}_i^2}{N \sum_{m'} f_{m'} R_{mm'}^2 + \sigma^2} + \log(N \sum_{m'} f_{m'} R_{mm'}^2 + \sigma^2)$$

which tries to fit $N \sum_{m'} f_{m'} R_{mm'}^2 + \sigma^2$ to $N\hat{\beta}_i^2$. This is exactly the idea of LD score regression (Eqn. 4). Therefore LDSR can be thought of as our objective in the extreme case assuming every $\hat{\beta}$ was generated independently.

## 4.2. Fast linear algebra with iterative algorithms

To calculate the likelihood, we must now invert and calculate the log determinant of $A_\theta^{(i)}$. However, this matrix is large, and often singular, making calculations expensive and unstable. We address these problems by rewriting the loss in terms of a well-conditioned matrix and then efficiently calculating its inverse using iterative algorithms.

### 4.2.1. Reformulating the loss

Previous works like Salehi Nowbandegani et al. (2023) or Hormozdiari et al. (2014), deal with the singularity issues by adding regularization to $R_{(i),(i)}$ as $R_{(i),(i)} + \epsilon I$ for some small $\epsilon$, which trades off numerical stability for bias. In contrast, we can work with the exact $R$ by using the Woodbury identity and the matrix determinant lemma to write the loss in terms of the pseudo-inverse $R_{(i),(i)}^\dagger$ (see Appendix C.1): we re-write $(A_\theta^{(i)})^{-1}$ as

$$(A_\theta^{(i)})^{-1} = \frac{1}{\sigma_N^2} R_{(i),(i)}^\dagger -$$

$$L_{(i)}^\intercal F_\theta^{(i)\frac{1}{2}} \left( I + \frac{1}{\sigma_N^2} F_\theta^{(i)\frac{1}{2}} W_{(i)} F_\theta^{(i)\frac{1}{2}} \right)^{-1} F_\theta^{(i)\frac{1}{2}} L_{(i)}$$

where $L_{(i)} = R_{(i)^+,(i)} R_{(i),(i)}^\dagger$ and $W_{(i)} = R_{(i)^+,(i)} R_{(i),(i)}^\dagger R_{(i),(i)^+}$, and $|A_\theta^{(i)}|$ as

$$\log |A_\theta^{(i)}| = \log |\sigma_N^2 R_{(i),(i)}|$$
$$+ \log |I + \frac{1}{\sigma_N^2} F_\theta^{(i)\frac{1}{2}} W_{(i)} F_\theta^{(i)\frac{1}{2}}|.$$

Note the matrices $R_{(i),(i)}^\dagger$, $L_{(i)}$, and $W_{(i)}$ do not depend on $\theta$ and can therefore be calculated before training begins.

Given the previous simplifications, now the only challenging linear algebra computation to perform at every step is to invert and calculate the log determinant of the matrix

$$B_\theta^{(i)} = I + \frac{1}{\sigma_N^2} F_\theta^{(i)\frac{1}{2}} W_{(i)} F_\theta^{(i)\frac{1}{2}}.$$

Although the dimensions of $B_\theta^{(i)}$ are the size of $(i)^+$ – possibly 3 times the size of $(i)$ – it is strictly positive definite and, since $\sigma_N^{-2} \times F_\theta \approx N \times M^{-1} < 1$, it is typically well-conditioned. This last fact means that despite its increased size, we can actually invert it substantially faster than $A_\theta^{(i)}$ using iterative algorithms.

### 4.2.2. Iterative algorithms

To invert and calculate the log determinant of $A_\theta^{(i)}$ or $B_\theta^{(i)}$, we could perform a Cholesky decomposition. Unfortunately, this is $O(M_i^3)$ where $M_i$ is the number of rows of the matrix; this is too computationally expensive to perform at each step and becomes more so if we wish to include more variants in our study in the future.

Luckily, for well-conditioned matrices, we can achieve much better computational complexity by using iterative methods like stochastic Lanczos quadrature (SLQ) (Golub & Loan, 2018; Saad, 2011) for $\log |B_\theta^{(i)}|$ and conjugate gradients (CG) (Nocedal & Wright, 2006; Golub & Loan, 2018; Saad, 2003) for solves $(B_\theta^{(i)})^{-1}$. Both methods rely on performing multiplications against $B_\theta^{(i)}$ at each iteration and each iteration typically exponentially improves the quality of the approximation, typically until machine precision. Thus, the computational cost of both methods is a manageable $\mathcal{O}(M_i^2 K)$ where $K$ is the number of iterations and $\mathcal{O}(M_i^2)$ is an upper bound on the cost of doing a matrix-multiply with $B_\theta^{(i)}$. The number of iterations required to converge below an error threshold of these iterative methods is directly linked to the eigenspectrum of the matrix (Nocedal & Wright, 2006; Saad, 2011; Hogben, 2013) – since $B_\theta^{(i)}$ is well conditioned, we expect it to need few iterations $K$ and therefore fewer computational resources.

To implement these methods, we use CoLA (Potapczynski et al., 2023), a numerical linear algebra library that is compatible with diverse deep learning frameworks and that provides backpropagation capabilities for SLQ and CG. CoLA

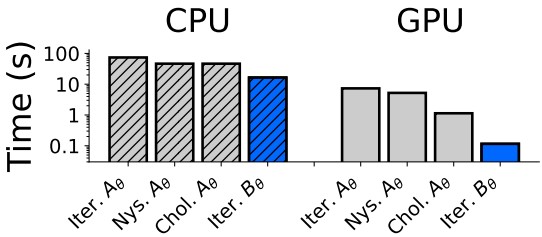

*Figure 2.* **DeepWAS efficiently computes the loss and its gradient.** We measure the time it takes to compute our loss Eqn. 3 as well as its gradients with respect to $\theta$. We do this for 20 mini-batches of real UKBB data and display the mean runtime as barplots. `Chol` stands for Cholesky decomposition. `Iter` stands for the iterative algorithms SLQ and CG. `Nys` stands for the iterative algorithms with Nyström preconditioning. We set a relative tolerance of $10^{-6}$ for CG and use 100 samples of SLQ yielding an average relative error on the gradient of 2.5%. For GPU we used an NVIDIA A100-SXM4-80GB and for CPU Intel(R) Xeon(R) Platinum 8268 CPU @ 2.90GHz.

computes the gradients of $B_\theta^{-1}$ and $\log |B_\theta^{-1}|$ by using the following identities

$$\nabla_\theta B_\theta^{-1} = -B_\theta^{-1} \nabla_\theta B_\theta B_\theta^{-1}$$
$$\nabla_\theta \log \left| B_\theta^{-1} \right| = \mathrm{trace}(B_\theta^{-1} \nabla_\theta B_\theta)$$
$$= \mathbb{E}_{u \sim \mathcal{N}(0,I)} (B_\theta^{-1} u)^\intercal \nabla_\theta B_\theta u$$

where both quantities require backpropagating through $B_\theta$ only and where we use the Hutchinson trace estimator. Additionally, `CoLA` allows us to leverage GPU acceleration for our numerical techniques which significantly reduces the runtime.

In Fig. 2 we compare the cost of computing the likelihood Eqn. 3 by using iterative algorithms on $A_\theta^{(i)}$ or $B_\theta^{(i)}$, or by performing Cholesky decomposition on $A_\theta^{(i)}$ (performing Cholesky on $B_\theta^{(i)}$ would take longer due to its increased number of rows). We also compare with Nyström preconditioning, a popular method for solving large systems in machine learning that effectively makes $A_\theta^{(i)}$ better conditioned (Gardner et al., 2018). We see that, despite being larger, we can invert $B_\theta^{(i)}$ faster than $A_\theta^{(i)}$ by leveraging its being well conditioned using iterative algorithms, even when using Nyström preconditioning. We also further accelerate our calculations by performing them on GPUs rather than CPUs, which had previously been used to train functionally informed priors.

# 5. Predicting variant effects from genomic features

Now we have a method for accurately and efficiently training a large model $f_\theta$. Here we specify how we build $f_\theta$ that include many more functional and comparative genomics

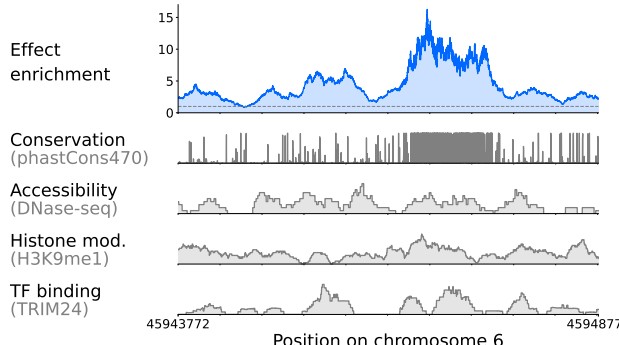

*Figure 3.* **DeepWAS flexibly predicts the effect of a variant on a phenotype from a variety of genomic features.** We plot the predicted effect enrichment $\sqrt{f_{\theta,m}/\mathbb{E}f_{\theta,m}}$ for hypothetical variants at all positions in a 5000-long non-coding window of chromosome 6, with the expectation over variants in UKBB. We compare with selected genomic features fed into $f_\theta$. We use predictions from a DeepWAS transformer-based model trained on height in UKBB.

features than previous works in Sec. 5.1. We describe large flexible neural network architectures for $f_\theta$ in Sec. 5.2. The result is a flexible function that predicts the effect of a variant from a variety of genomic information (Fig. 3).

## 5.1. Features

Previous methods have built $f_\theta$ using functional genomics features such as DNA accessibility, proximity to functional elements, and presence of a coding region and comparative genomics features such as conservation scores (Finucane et al., 2015; Li et al., 2024). Many of these features are defined as annotations at each position in the genome; to get a single value, annotations were averaged in a window before being passed to $f_\theta$.

We expand this set in two ways. First we consider a significantly expanded set of functional genomics annotations – binding and accessibility annotations from ENCODE (ENCODE Project Consortium, 2012), enhancer annotations from FANTOM (Lizio et al., 2015) – and comparative genomics annotations – conservation scores such as PhyloP (Pollard et al., 2010), and predictions of effects of mutations in coding regions such as those from ESM2 (Liu et al., 2020). Details of these data are in App. D.2 and D.3. Second, instead of considering an average of the values of annotations in a window around the variant, we pass the model the exact values of the annotations at all positions in the window.

Gazal et al. (2017) used LDSR to demonstrate that the recent history of a variant in humans can also be predictive of its effect size. To account for this, we also included the frequency of each variant $m$, $\mathrm{freq}_m$; its "minor allele" frequency,

$\min\{\text{freq}_m, 1 - \text{freq}_m\}$; and its LD score $\sum_{m'} R_{mm'}^2$ as features.

## 5.2. Architecture

For all genomics annotations but coding mutation effect predictions, we consider a window around each variant of size $w$. We pass these functional annotations $C_{\text{func},m} \in \mathbb{R}^{d_{\text{func}} \times w}$ along with predictions of the effects of mutations if the mutation is in a coding region and genomic architecture information, $C_{\text{pred},m} \in \mathbb{R}^{d_{\text{pred}}}$, to a neural network $f_\theta(C_{\text{func},m}, C_{\text{pred},m})$. In our case, $d_{\text{func}} = 165$, and $d_{\text{pred}} = 9$; we also choose a window size of $w = 256$. We use a transformer-CNN hybrid architecture adapted from a network used to predict functional annotations from sequence, Enformer (Avsec et al., 2021)[4]; this architecture uses a mix of convolutional and attention layers. We can change the dimension of the internal representation to alter the number of parameters of the model.

Speed et al. (2017) suggested that setting $f = \text{constant}$ in our model makes the implicit assumption that rare variants have larger effects. They remove this assumption with a more general model $f_m = (\text{freq}_m(1 - \text{freq}_m))^\alpha$ where $\alpha$ is a fit parameter. In our case, we consider

$$f_{\theta,m} = (\text{freq}_m(1 - \text{freq}_m))^\alpha \text{NN}_\theta(C_{\text{func},m}, C_{\text{pred},m}) \quad (6)$$

where $\text{NN}_\theta$ is a neural network or any other model. In many of our experiments, $\alpha$ typically converged to a value between 0.6 and 0.7 regardless of its initialization; to simplify our model comparisons below, we fixed it at 0.7 for all experiments.

## 6. Empirical Results

We now apply DeepWAS to train flexible neural network models in order to better explain which variants are associated with phenotypic traits. See A for experimental details.

### 6.1. DeepWAS learns true $f$ from semi-synthetic data

We first wish to check if our gradient descent procedure can recover the true $f$ from noisy genetic data. Unfortunately we do not have access to real $f$ in real data. We therefor turn to semi-synthetic simulations, which are a staple of statistical genetics literature (ex. Candès et al. (2016) or O'Connor et al. (2019)), used to validate methods when true effects $\beta$ are unavailable.

We use real LD patterns $R$ from public data from the UK Biobank (UKBB) study of over 300,000 individuals of Eu-

---
[4]We do not use pretrained weights as the original Enformer architecture regressed from sequence to predict functional annotations, rather than our task which is to take in functional annotations to predict effect on phenotype.

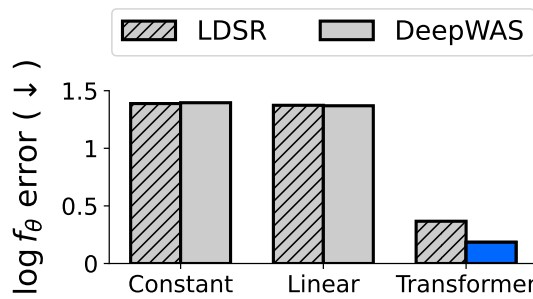

*Figure 4.* **DeepWAS using a transformer model best recovers the true $f$.** The bars represent the RMSE difference between the learnt $f_\theta$ and the ground truth $f$ in the log space evaluated over a set of validation functional annotations.

ropean ancestry (Weissbrod et al., 2020) (see App. D.1). We generate synthetic $\hat{\beta}$ as follows:

$$\beta_m \sim \mathcal{N}\left(0, f_m\left(C_{\text{func},m}, C_{\text{pred},m}\right)\right); \hat{\beta} \sim \mathcal{N}\left(R\beta, \sigma_N R\right)$$

where $f$ represents that function that we are trying to learn. We consider $f_\theta$ as a randomly initialized transformer-based neural network model (52 million parameters). See App. A.5 for details.

We tried fitting this data with models based on Eqn. 6. We used simple models – $\text{NN}_\theta = \text{constant}$ and $\text{NN}_\theta = \text{gen}$-eralized linear model (see App. A.2) – and a more flexible $\text{NN}_\theta$ with the transformer architecture with LD score regression (LDSR) and DeepWAS. In Figure 4 we show DeepWAS with a large model can closely recover the true variant effect distribution $f_\theta$ – it achieves a low error in predictions $f_\theta$. Furthermore, this model better predictions than restricted constant and linear $f_\theta$. We also see that our method makes more accurate predictions than models trained with LD score regression. In Sec. B.1 we also consider inferring a more challenging ground truth $f$.

### 6.2. Predicting phenotype on UK Biobank

Now we attempt to better explain phenotypes $y$ of three traits from UK Biobank calculated in Loh et al. (2015) (see App. D.1) – body mass index, height, and asthma. Unfortunately, we only have access to the public $\hat{\beta}$ rather than the private $y$. Using just public data we can however hold out some chromosomes and evaluate our models on the following task:

- train a model $\theta$ on phenotypes $y$ and genotypes $X$ of held-in chromosomes,

- then predict phenotypes $y$ using only genotypes $X$ of held-out chromosomes.

- We report how much the marginal likelihood (Eqn. 1) increases using the model on held-out chromosomes

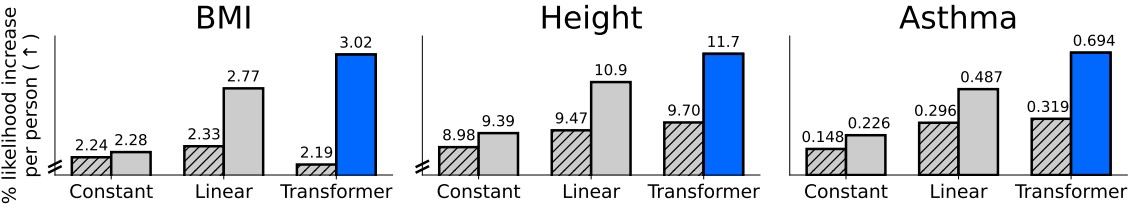

*Figure 5.* **More flexible models trained with DeepWAS better explain phenotype using held-out chromosomes.** We try to predict the phenotype $y$ of individuals in UK Biobank using only variants in held out chromosomes 6, 7, 8 as in Eqn. 1. We report how much the marginal likelihoods increases when we use different priors $F$, compared to a model which assumes there is no genetic effect on the phenotype ($F = 0$). When training with DeepWAS, larger models better fit the data. While larger models perform worse when training with LDSR. Legend is identical to Fig. 4.

compared to a "null" model that assumes there is no genetic effect on $y$ ($f = 0$). We then divide the marginal log likelihood by $N$ and exponentiate to get a quantity "per person".

We show in App. C.2 that this is equivalent to training our model using public association data $\hat{\beta}$ on held-in chromosomes and reporting the marginal likelihood (Eqn. 3) of association data $\hat{\beta}$ on held-out chromosomes. In our experiments, we hold out chromosomes 6, 7, and 8 and train on all other autosomal chromosomes.

**Using flexible models with DeepWAS results in better explanations of phenotype data.** We evaluate the ability of various architectures trained with LDSR and DeepWAS to predict phentotype using held-out chromosomes. LDSR and DeepWAS used similar computational resources in training.

Fig. 5 shows that a model with constant $f$ better explains the data than a null model, $f = 0$ – this represents how much better our prediction of phenotype is simply from including information about genotype $X$ in our model. We next see that including functional annotations in a linear prior increases prediction quality – this represents how much better our prediction is from including any information about genomic features. The varying improvements across the phenotypes correlate with their reported "SNP-heritabilities" (Hou et al., 2019) – more than 50% of the variation of height seen in the population can be explained by variants in our study, while that number is roughly 30% for BMI and 10-16% for asthma.

We next increase the model size to a transformer. Surprisingly, when training using LDSR, the quality of prediction does not significantly increase, and in the case of BMI, even decreases. The more flexible architecture potentially overfits the data due to the loss of statistical efficiency when performing LDSR. When accounting for correlations in the $\hat{\beta}$ with DeepWAS however, the larger transformer substantially outperform all other models.

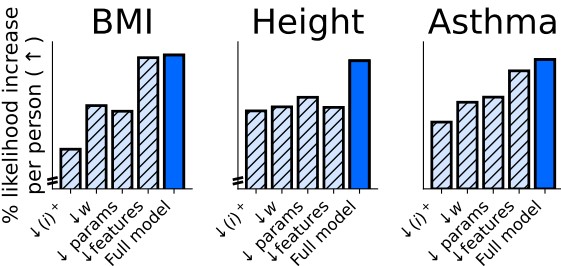

*Figure 6.* **Ablations show that larger models with more features better explain genetic associations.** We ablate the number of variants in an LD window ($(i)^+$), the feature set ($w$ and the presence of conservation features) and the model size of our transformer-based model. We report statistics as in Fig. 5, with the same y-axes. We compare them to the full model (dark blue) reporting the same value as in Fig. 5. For exact numbers see App. B.2.

**Larger features and larger models improve prediction.** We finally set out to determine the importance of the number of variants we trained on, the feature set, and model size for performance. We trained models with fewer variants per window – we reduced the size of $(i)^+$ relative to $(i)$ from windows of 1'000'000 to 100'000 (see Sec. 4.1) – a reduced feature set – we looked at a window of $w = 16$ rather than $w = 256$ around each variant, and removed the conservation features from phastCons and PhyloP (see Sec. 5.1) – and with a reduced size – we reduce the number of parameters from 52 million to 3.9 million. We chose to ablate the conservation features as these have been seen to be particularly important for effect size prediction in previous work (Finucane et al., 2015).

Fig. 6 shows that ablating the number of variants in each window, the model size, or feature set often harms model performance. This shows that saving compute by considering smaller $(i)^+$ severely harms performance. This also suggests that our model benefits from its flexibility and features to better predict the effects of variants.

# 7. Discussion

By efficiently inverting LD matrices, DeepWAS allows us to train large models that better predict the effects of variants on phenotype and to learn their functional causes. Our results demonstrate that larger models make better predictions than the simple models used in practice, and that increasing the model size and using more features improves predictive power. Our work suggests that even larger models trained on more data may result in further improvements in the future.

**Confounding and other priors**   Future work may address some of the limitations of DeepWAS. First, LD score regression can deal with some confounding from population stratification (Bulik-Sullivan et al., 2015), while such effects were only accounted for in DeepWAS by using population-covariate-adjusted BOLT-LMM public statistics (Loh et al., 2015). Future work may investigate the effectiveness of this method. Next, DeepWAS considers a simple linear model with a single component normal prior. More flexible models with multi-component priors or accounting for non-linearity would be a natural next step (Loh et al., 2015; Zhang et al., 2021).

**Larger models on more data**   DeepWAS could be expanded to include other sources of information, such as embeddings of genes that are near variants (Weeks et al., 2020; Theodoris et al., 2023) and use pre-trained models (Avsec et al., 2021; Dalla-Torre et al., 2023). In this work, we only looked at a single European population; for more robust estimates DeepWAS could be trained to learn across populations (Spence et al., 2022). Finally, rather than training on a single phenotype at a time, DeepWAS could learn from multiple phenotypes (Morgante et al., 2023) boosting power to learn common patterns.

**Boosting signal in rare or low-heritability disease**   We saw in Fig. 5 that the flexibility of a larger model made the largest impact in asthma, where the "SNP-heritibility" was low – since signal-to-noise was lower, an improved prior made a larger impact. DeepWAS could potentially impact other settings with low signal-to-noise, such as the interpretation of rare variants (Lee et al., 2014), or, using pre-trained DeepWAS models, the analysis of small cohorts (Huang et al., 2024) or rare disease (Samocha et al., 2014).

As well, inference in low signal-to-noise settings in genetics typically require bespoke tools and analyses. Using an empirical Bayesian framework to account for uncertainty, Deep-WAS can both leverage functional insights from common variants and extract signal from rare variants themselves, providing a unified framework across the allele frequency spectrum.

## Acknowledgments

We thank Benjamin Strober and David Knowles for helpful advice on dealing with the idiosyncrasies of genetic data and literature. We thank Shenglong Wang at NYU IT High Performance Computing for help optimizing the data loader to work efficiently on high performance compute resources. This work was supported in part by NSF CAREER IIS-2145492, NSF CDS&E- MSS 2134216, NSF HDR-2118310, BigHat Biosciences, Capital One, and an Amazon Research Award.

## Impact Statement

This paper presents work whose goal is to leverage large data to better predict the genetic causes of human traits. It could be used to better predict disease in the future. On the other hand, genetics data is not collected uniformly across the population; therefore building models using existing large data may only improve prediction for subsets of the population, exacerbating health outcome inequalities.

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

# A. Experimental details

## A.1. Models

We obtained code for enformer from https://github.com/lucidrains/enformer-pytorch under the MIT license. We set the internal dimension to 1536 and the number of transformer layers to 2. Our "smaller model" in the ablations reduced the internal dimension to 384.

We normalize features to have mean 0 and variance 1 across the genome before passing them to any model.

## A.2. Generalized linear model

As a baseline we consider a generalized linear model as suggested in Li et al. (2024) using averages of each functional annotation in the window as in Finucane et al. (2015) (although we only consider a single window size):

$$f_{\theta,m} = (\text{freq}_m(1 - \text{freq}_m))^\alpha \exp\left(\sum_d w_d \sum_w C_{\text{func},m,d,w} + \sum_{d'} w'_{d'} C_{\text{pred},m,d'} + c\right)$$

where $(w_d)_d$, $(w'_{d'})_{d'}$, and $c$ are learnable parameters.

## A.3. Training

We trained our models with an AdamW optimizer with default hyperparameters, 100 warmup steps with a linear schedule. We use a learning rate of 0.0001 for transformer models and 0.001 for linear and constant models. We train all models for 10 epochs. We train models on single A100 GPUS on an academic cluster; transformer-based models were trained for 15 to 20 hours.

For all models we use $\sigma$ calculated using BOLT-LMM from Loh et al. (2015).

For all models, we use 12 data loader workers in a PyTorch (Paszke et al., 2019) `Dataloader` object. To most efficiently use IO capacity, each loader loads 100 batches linearly from the same chromosome before switching to a different location. To average across this bias, we accumulate gradient steps across 12 steps.

For the reduced LD window ablation, for each original $(i)^+$ window of size 3'000'000, we set $(i)^+$ as a random window of up to 6'000 contiguous variants – we chose 6'000 as it is near the average size of $(i)$ in our standard approach. Then we set $(i)$ as all the variants further than 100'000 from the ends of $(i)^+$.

## A.4. LD score regression (LDSR)

Finucane et al. (2015) suggested performing the linear LD score regression with a square loss 1) dividing by the (rough) standard deviation of a chi-squared variable and 2) downweighting variants in LD with many other variants: calling $l = \mathbb{1}^T R^{\circ 2}$ and $h_g^2 = E_i f_{\theta,i}$ (ballpark estimate made before training), we minimize (we also multiply numerator and denominator by $N$)

$$\sum \frac{1}{l_i} \frac{1}{(Nh_g^2 l_i/M + 1)^2} \left(\frac{N}{M} R_i^{\circ 2} f_\theta + \sigma^2 - N\hat{\beta}_i^2\right)^2.$$

## A.5. Simulation

Here we describe how we chose a realistic $f$ for semi-synthetic simulation. Recall,

$$y \sim N(0, \frac{1}{N}X^T F X + \sigma^2 I).$$

Therefore

$$1 = \text{Var}(y_{i,i}) = \sigma^2 + \frac{1}{N}\sum_m X_{m,i}^2 F_m.$$

Assuming presence of a variant $X_{m,i}$ is independent of $F_m$, we have

$$1 = \text{Var}(y_{i,i}) \approx \sigma^2 + \frac{M}{N}E_m[X_{m,i}^2]E_m[F_m] = \sigma^2 + \frac{M}{N}E_m[F_m].$$

Thus, in our simulated data, ideally we would ensure that

$$E_m F_m = \frac{N}{M}(1 - \sigma^2).$$

In our case, we choose a highly heritable disease with $\sigma^2 = 1/2$ so half of the variance of $y$ is from the noise $\epsilon$ and the other half is genetic. Using real values $N = 407527$ and $M = 11904924$ for our data, we set $E_m f_m = \frac{N}{2M}$ by initializing a $\tilde{f}$, calculating $E_m \tilde{f}_m$, and defining $f_m = \frac{N}{2M E_m \tilde{f}_m} \tilde{f}_m$.

We defined $\tilde{f}_m = \exp(10 \times \mathrm{NN}_\theta(C_{\mathrm{func},m}, C_{\mathrm{pred},m}))$ where $\mathrm{NN}_\theta$ is a randomly initialized Enformer model.

## B. Additional results

### B.1. Biologically realistic ground truth for semi-synthetic experiment

In Sec. 6 we considered a semi-synthetic setting where the ground truth $f$ was a transformer-based model. In this case, when our model class is well-specified, we recovered the true $f$. Here, we consider a biologically-motivated ground truth $f$ which is designed to be challenging – we are interested in measuring if there is still a benefit to using a more flexible model if our model class is misspecified.

We consider a function which adds an effect enrichment whenever the sum of functional annotations in a window is above a threshold. Let $C = [C_{\mathrm{func}}, C_{\mathrm{pred}}]$ be the concatenation for $C_{\mathrm{func}}$ and $C_{\mathrm{pred}}$ in the same order they're presented in App. D, with $C_{\mathrm{pred}}$ broadcasted to the same window size as $C_{\mathrm{func}}$. Thus, the ground truth $f$ takes the shape

$$\log f(C) = \sum_{d \in \{0,2,7,12\}} v_d \mathbb{1}\left(\sum_{w \in \{113,\dots,143\}} C_{d,w} > e_d\right) - \log M$$

where $v_0 = 0.7$, $v_2 = 0.5$, $v_7 = 1.37$, $v_{12} = 0.5$ and $e_0 = 0$, $e_2 = 0$, $e_7 = -20$ and $e_{12} = -10$. This is a challenging $f$ for our transformer-based model because the signal is sparse (depending only on 4 features), discontinuous, multi-modal and it only depends on 32 positions at the center of the window of size 256. We chose the values of $e_d$ so that feature $d = 0$ is active about $\approx 50\%$ of the time across the genome, $d = 2$ about $\approx 25\%$, $d = 7$ always active and $d = 12$ about $\approx 50\%$. Then, we set the values of $v_d$ so that $f$ is multi-modal with values ranging from 0.072 up to 0.40.

In Fig. 7 we see, even when misspecified, using a more flexible model class leads to a better fit of the true function.

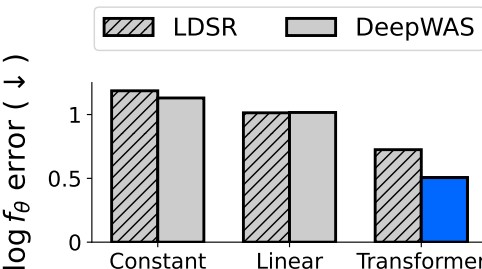

*Figure 7.* **DeepWAS using a transformer model best recovers the true $f$ even in a challenging setting.** The bars represent the RMSE difference between the learnt $f_\theta$ and the ground truth $f$ in the log space evaluated over a set of validation functional annotations.

### B.2. Ablation results in a table

We present the numerical results of Fig. 6 in Tab. 1.

*Table 1.* Model Ablation Results: Percentage Likelihood Increase per Person

| Model Configuration | BMI | Height | Asthma |
|---|---|---|---|
| Reduced LD window $(i)^+$ | 2.38 | 10.32 | 0.358 |
| Reduced Feature window $w$ | 2.68 | 10.43 | 0.465 |
| Reduced Parameters (reduce hidden dimension) | 2.64 | 10.69 | 0.492 |
| Reduced Features (remove 15 conservation features) | 3.01 | 10.42 | 0.633 |
| **Full Model** | **3.02** | **11.70** | **0.694** |

# C. Derivations

## C.1. Loss derivation

In its more general form, the Woodbury matrix identity (also known as the Sherman-Morrison-Woodbury formula) states that $(A + UCV)^{-1} = A^{-1} - A^{-1}U(C^{-1} + VA^{-1}U)^{-1}VA^{-1}$ and the matrix determinant lemma (also known as the Weinstein-Aronszajn identity) states that $|A + UCV| = |C||A||C^{-1} + VA^{-1}U|$.

Recall that $A_\theta^{(i)} = R_{(i),(i)^+}F_\theta^{(i)}R_{(i)^+,(i)} + \sigma_N^2 R_{(i),(i)}$. Applying Woodbury to $(A_\theta^{(i)})^{-1}$ and defining $L^{(i)} = R_{(i)^+,(i)}R^\dagger$ and $W^{(i)} = R_{(i)^+,(i)}R_{(i),(i)}^\dagger R_{(i),(i)^+}$, we get

$$
\begin{aligned}
(A_\theta^{(i)})^{-1} =& \frac{1}{\sigma_N^2}R_{(i),(i)}^\dagger - \frac{1}{\sigma_N^2}R_{(i),(i)}^\dagger R_{(i),(i)^+}((F_\theta^{(i)})^{-1} + \frac{1}{\sigma_N^2}R_{(i)^+,(i)}R_{(i),(i)}^\dagger R_{(i),(i)^+})^{-1}\frac{1}{\sigma_N^2}R_{(i)^+,(i)}R_{(i),(i)}^\dagger \\
=& \frac{1}{\sigma_N^2}R_{(i),(i)}^\dagger - \frac{1}{\sigma_N^2}L^{(i)\intercal}F_\theta^{(i)\frac{1}{2}}(I + \frac{1}{\sigma_N^2}F_\theta^{(i)\frac{1}{2}}W^{(i)}F_\theta^{(i)\frac{1}{2}})^{-1}\frac{1}{\sigma_N^2}F_\theta^{(i)\frac{1}{2}}L^{(i)} \\
=& \frac{1}{\sigma_N^2}R_{(i),(i)}^\dagger - \frac{1}{\sigma_N^2}L^{(i)\intercal}F_\theta^{(i)\frac{1}{2}}(B_\theta^{(i)})^{-1}\frac{1}{\sigma_N^2}F_\theta^{(i)\frac{1}{2}}L^{(i)}.
\end{aligned}
$$

Moreover, for $|A_\theta^{(i)}|$ we have

$$
\begin{aligned}
|A_\theta^{(i)}| &= |\sigma_N^2 R_{(i),(i)}|_+|F_\theta^{(i)}||(F_\theta^{(i)})^{-1} + \frac{1}{\sigma_N^2}W^{(i)}| \\
&= |\sigma_N^2 R_{(i),(i)}|_+|F_\theta^{(i)}||F_\theta^{(i)\frac{1}{2}}|^{-1}|I + \frac{1}{\sigma_N^2}F_\theta^{(i)\frac{1}{2}}W^{(i)}F_\theta^{(i)\frac{1}{2}}||F_\theta^{(i)\frac{1}{2}}|^{-1} \\
&= |\sigma_N^2 R_{(i),(i)}|_+|B_\theta^{(i)}|,
\end{aligned}
$$

where $|\cdot|_+$ stands for the pseudo-determinants. As we discussed before, $R_{(i),(i)}$ should be strictly positive-definite but, due to numerical precision, $R_{(i),(i)}$ could contain eigenvalues close to zero, or negative. This is why we use $R_{(i),(i)}^\dagger$ and why we would only consider de positive eigenvalues for the computation of $|\sigma_N^2 R_{(i),(i)}|_+$.

## C.2. Held-out chromosome prediction using public data

We essentially show that, since Eqn. 2 is derived from Eqn. 1, their likelihoods can only differ by a constant.

The negative log likelihood of $y$ using only held-in chromosomes $X_{\text{in}}$ (Eqn. 1) minus the log likelihood of the null ($F = 0$) is

$$
\frac{1}{2}\left[y^\intercal(X_{\text{in}}^\intercal F_{\text{in}}X_{\text{in}} + \sigma^2 I)^{-1}y - y^\intercal(\sigma^2 I)^{-1}y + \log|X_{\text{in}}^\intercal F_{\text{in}}X_{\text{in}} + \sigma^2 I| - \log|\sigma^2 I|\right] \tag{7}
$$

while the same equation for $\hat{\beta}$ (Eqn. 2) is, defining pseudo-determinants $|\cdot|_+$,

$$
\frac{1}{2}\left[\hat{\beta}^\intercal(R_{\text{in}}F_{\text{in}}R_{\text{in}} + \sigma_N^2 R_{\text{in}})^\dagger\hat{\beta} - \hat{\beta}^\intercal(\sigma_N^2 R_{\text{in}})^\dagger\hat{\beta} + \log|R_{\text{in}}F_{\text{in}}R_{\text{in}} + \sigma_N^2 R_{\text{in}}|_+ - \log|\sigma_N^2 R_{\text{in}}|_+\right]. \tag{8}
$$

By Woodbury, the first two terms Eqn. 7 are equal to

$$
\frac{1}{2}\left[-\frac{1}{\sigma^4}y^\intercal X_{\text{in}}^\intercal(F_{\text{in}}^{-1}+\sigma^{-2}X_{\text{in}}^\intercal X_{\text{in}})^{-1}X_{\text{in}}y\right]=-\frac{1}{2\sigma_N^2}\hat\beta^\intercal(\sigma_N^2 F_{\text{in}}^{-1}+R_{\text{in}})^{-1}\hat\beta
$$
$$
=-\frac{1}{2\sigma_N^2}\hat\beta^\intercal(\sigma_N^2 F_{\text{in}}^{-1}+R_{\text{in}}R_{\text{in}}^\dagger R_{\text{in}})^{-1}\hat\beta
$$
$$
=\frac{1}{2\sigma_N^2}\hat\beta^\intercal(R_{\text{in}}\sigma_N^{-2}F_{\text{in}}R_{\text{in}}+R_{\text{in}})^\dagger\hat\beta-\frac{1}{2\sigma_N^2}\hat\beta^\intercal R_{\text{in}}^\dagger\hat\beta
$$

which is equal to the first two terms of Eqn. 8. Meanwhile by a similar application of the matrix determinant lemma, the last two terms Eqn. 7 are equal to the last two terms of Eqn. 8. The same is true replacing $X_{\text{in}}$ with only held out chromosomes $X_{\text{out}}$.

## D. Data Collection

### D.1. UKBB public statistics

We downloaded UK biobank LD matrices computed in Weissbrod et al. (2020) from the Amazon web services S3 container `s3://broad-alkesgroup-ukbb-ld/UKBB_LD/`. These matrices can have small negative eigenvalues, which we removed prior to training. We also, as is standard, do not include regions with long-range LD in chromosomes 6, 8, and 11.

We downloaded UK biobank association statistics computed using BOLT-LMM (Loh et al., 2015) from the UKBB_409K folder in `https://console.cloud.google.com/storage/browser/broad-alkesgroup-public-requester-pays`. These association statistics also contained frequencies of each variant. Any variants that have LD information but that are missing associations are discarded; all variants with association information also had LD information.

UKBB coordinates are in GrCh37 but many of our features below are in the GrCh38 build. We used rsid's and pyliftOver (`https://github.com/konstantint/pyliftover`) to map to GrCh38. For the handful of variants we could not map we gave them the location of a nearby variant.

### D.2. Coding variant annotations

We downloaded the predictions of the effects of variants in coding regions from various models from `https://www.dbnsfp.org/`. We used six predictions labeled `ESM1b_score`, `GERP++_RS`, `SIFT_score`, `PROVEAN_score`, `FATHMM_score`, `EVE_score`. For non-coding variants or variants missing a prediction, we set $C = 0$.

### D.3. Functional and conservation annotation data

**Conservation** We downloaded bigWig files of our phylogenetic correlation annotations from `http://hgdownload.soe.ucsc.edu/goldenPath/hg38/` (Pollard et al., 2010; Hubisz et al., 2011). We used 15 PhyloP and phastCons scores made from various alignments: `phyloP470way`, `phyloP447way`, `phyloP100way`, `phyloP30way`, `phyloP20way`, `phyloP17way`, `phyloP7way`, `phyloP4way`, `phastCons470way`, `phastCons100way`, `phastCons30way`, `phastCons20way`, `phastCons17way`, `phastCons7way`, `phastCons4way`.

**FANTOM** We downloaded hCAGE FANTOM annotations of human tissues from `https://fantom.gsc.riken.jp/5/datahub/hg38/tpm/human.tissue.hCAGE/` (Lizio et al., 2015). This gave us roughly 400 annotations; we picked a random 20 tissues from this set and collected forward and backward CAGE annotations for each tissue, giving us a total of 40 features. The tissues were `lymph node, adult, donor1; heart, adult, diseased post-infarction, donor1; skeletal muscle, adult, pool1; occipital lobe, adult, donor1; parietal cortex, adult, donor10258; thymus, adult, pool1; thyroid, adult, pool1; pons, adult, pool1; parotid gland, adult; Fingernail (including nail plate, eponychium and hyponychium), donor2; thalamus, adult, donor10258; caudate nucleus, adult, donor10252; parietal lobe, adult, donor10252; cerebrospinal fluid, donor2; kidney, fetal, pool1; eye - muscle`

```
inferior rectus, donor1; nucleus accumbens, adult, pool1; parietal lobe – adult,
donor10196; cerebral meninges, adult; throat, adult.
```

**ENCODE**  We downloaded bigWig files of functional genomics annotations from https://www.encodeproject.org/search/ (ENCODE Project Consortium, 2012). We did not use annotations with warnings, errors, or that were non-compliant. We used assays with titles `TF ChIP-seq`, `Histone ChIP-seq`, `eCLIP`, `total RNA-seq`, `polyA plus RNA-seq`, `polyA minus RNA-seq`, `small RNA-seq`, `microRNA-seq`, `ChIA-PET`, `WGBS`, `DNase-seq`, `ATAC-seq`, `PRO-cap`, `PRO-seq`, `Bru-seq`, `BruChase-seq`, `RAMPAGE`, `PAS-seq` and those that had available bigWig files for GrCh38. We got over 100 `eCLIP` annotations of RNA binding; since each of these annotations are sparse, we summed them together to create a single `all_eCLIP` annotation. For `TF ChIP-seq` experiments that targeted a transcription factor, we only used assays from the 24 targets that had measurements from two or more labs.

Each of these experiments had multiple data annotations. We used the `fold_change_over_control` for a random replicate if it was available, otherwise we used a randomly chosen annotation. In total we had 91 annotations from ENCODE; the full list with bioproject ids is as follows: `all_eCLIP`, `polyA_minus_RNA-seq` (ENCSR000CQE), `TF_ChIP-seq of MTA3` (ENCSR391KQC), `Histone_ChIP-seq of H3K4me3` (ENCSR393ZOI), `TF_ChIP-seq of CREB1` (ENCSR897JAS), `TF_ChIP-seq of MCM3` (ENCSR990AZC), `TF_ChIP-seq of POLR2AphosphoS5` (ENCSR000BTW), `TF_ChIP-seq of NFIB` (ENCSR702BYX), `TF_ChIP-seq of SUZ12` (ENCSR757EMK), `TF_ChIP-seq of CAMTA2` (ENCSR336GFK), `TF_ChIP-seq of NFRKB` (ENCSR145BHD), `Histone_ChIP-seq of H3K36me3` (ENCSR524VDR), `TF_ChIP-seq of SIN3A` (ENCSR468LUO), `TF_ChIP-seq of PKNOX1` (ENCSR233FAG), `TF_ChIP-seq of NR3C1` (ENCSR516IVY), `TF_ChIP-seq of HLTF` (ENCSR090JNM), `Histone_ChIP-seq of H3K27me3` (ENCSR660TLD), `TF_ChIP-seq of NONO` (ENCSR476BQA), `TF_ChIP-seq of CBX8` (ENCSR616MOB), `TF_ChIP-seq of MCM7` (ENCSR068VRA), `TF_ChIP-seq of JUN` (ENCSR048CVK), `Histone_ChIP-seq of H3K23ac` (ENCSR592JNN), `TF_ChIP-seq of Cebpa` (ENCSR827TOM), `small_RNA-seq` (ENCSR000CRR), `TF_ChIP-seq of ZFX` (ENCSR027UFT), `TF_ChIP-seq of JUND` (ENCSR000BSA), `TF_ChIP-seq of MLX` (ENCSR873LYH), `TF_ChIP-seq of HDGF` (ENCSR200CUA), `TF_ChIP-seq of GATAD2A` (ENCSR160QYK), `PAS-seq` (ENCSR233JPT), `WGBS` (ENCSR999CXD), `RAMPAGE` (ENCSR413FKS), `TF_ChIP-seq of HDAC1` (ENCSR711VWL), `TF_ChIP-seq of DPF2` (ENCSR715CCR), `Histone_ChIP-seq of H2AFZ` (ENCSR859FGW), `TF_ChIP-seq of CTBP1` (ENCSR636EYA), `TF_ChIP-seq of SMARCA5` (ENCSR895HSJ), `TF_ChIP-seq of MNT` (ENCSR460XGV), `TF_ChIP-seq of BCOR` (ENCSR808AKZ), `TF_ChIP-seq of GTF2F1` (ENCSR557JTZ), `Histone_ChIP-seq of H3K56ac` (ENCSR036NSK), `Histone_ChIP-seq of H3K9ac` (ENCSR192IRQ), `TF_ChIP-seq of JUNB` (ENCSR431LRW), `Histone_ChIP-seq of H4K20me1` (ENCSR258TUP), `TF_ChIP-seq of TRIM24` (ENCSR957LDM), `TF_ChIP-seq of PLRG1` (ENCSR019KPC), `TF_ChIP-seq of FOXK2` (ENCSR171FUX), `Histone_ChIP-seq of H3K9me3` (ENCSR444YIP), `TF_ChIP-seq of GATAD2B` (ENCSR389BLX), `Histone_ChIP-seq of H2BK15ac` (ENCSR739BZR), `TF_ChIP-seq of KHSRP` (ENCSR686EYO), `TF_ChIP-seq of LARP7` (ENCSR288MOZ), `PRO-cap` (ENCSR100LIJ), `TF_ChIP-seq of SMARCA4` (ENCSR587OQL), `TF_ChIP-seq of PHB` (ENCSR650AWW), `ChIA-PET of POLR2A` (ENCSR217TFN), `Histone_ChIP-seq of H4K5ac` (ENCSR035BZI), `TF_ChIP-seq of CSDE1` (ENCSR626QJQ), `Bru-seq` (ENCSR849FAX), `TF_ChIP-seq of DMAP1` (ENCSR670YPQ), `microRNA-seq` (ENCSR934NMC), `ATAC-seq` (ENCSR265ZXX), `TF_ChIP-seq of ARNT` (ENCSR029IBC), `ChIA-PET of CTCF` (ENCSR030KAB), `TF_ChIP-seq of CEBPB` (ENCSR000BUB), `TF_ChIP-seq of RFXANK` (ENCSR823ADL), `TF_ChIP-seq of NBN` (ENCSR210ZYL), `TF_ChIP-seq of MLLT1` (ENCSR675LRO), `TF_ChIP-seq of HDAC2` (ENCSR330OEO), `TF_ChIP-seq of SP1` (ENCSR906PEI), `TF_ChIP-seq of RBBP5` (ENCSR330EXS), `TF_ChIP-seq of MTA2` (ENCSR551ZDZ), `TF_ChIP-seq of PBX3` (ENCSR000BTN), `Histone_ChIP-seq of H3K4me2` (ENCSR251DKX), `TF_ChIP-seq of POLR2A` (ENCSR388QZF), `TF_ChIP-seq of CBFA2T3` (ENCSR697YLJ), `TF_ChIP-seq of MAX` (ENCSR000BSH), `TF_ChIP-seq of YBX3` (ENCSR567JEU), `TF_ChIP-seq of RAD21` (ENCSR000BUC), `Histone_ChIP-seq of H3K79me1` (ENCSR213JMO), `DNase-seq` (ENCSR366NBE), `TF_ChIP-seq of CTCF` (ENCSR817HTJ), `Histone_ChIP-seq of H3K4me1` (ENCSR874NDX), `TF_ChIP-seq of TARDBP` (ENCSR801SWX),

TF_ChIP-seq of FOXP1 (ENCSR369YUK), TF_ChIP-seq of IKZF1 (ENCSR278JQG), PRO-seq (ENCSR988BZM), TF_ChIP-seq of ZBTB1 (ENCSR309ELI), TF_ChIP-seq of NCOA3 (ENCSR573OJP), Histone_ChIP-seq of H2BK20ac (ENCSR462XRE), TF_ChIP-seq of SUPT5H (ENCSR894CGX), Histone_ChIP-seq of H3K27ac (ENCSR010OVI), BruChase-seq (ENCSR672HUM), TF_ChIP-seq of EP300 (ENCSR686BQM).

