# OpenReview forum: "Training Flexible Models of Genetic Variant Effects from Functional Annotations using Accelerated Linear Algebra"
_ICML.cc/2025/Conference — ICML 2025 poster_

### Official Review · Reviewer_crZx · 2025-03-09

**Overall Recommendation:** 2

**Summary:**

This paper proposed WASP for phenotype prediction based on the genomic data, which utilizes an iterative algorithm and an approximation inverse to optimize the goal efficiently.  Experimental results on GWAS show its superiority over LD Score Regression.

**Claims And Evidence:**

Not very clear
1. The introduction notes that researchers in phenotype prediction typically emphasize incorporating functionally informed priors. However, my understanding of this paper is that its primary focus is on improving the optimization efficiency of e.q. (2) and (3). I did not observe any explicit methods for integrating or modeling such priors.
2. The paper mentioned that LD Score Regression is the SOTA method, which was published around 10 years ago. In the related work, there are some improved approximation methods in line 212 and some methods that utilize these priors in line 179. Did they improve the performance for phenotype prediction?

**Essential References Not Discussed:**

No

**Ethical Review Flag:**

Flag this paper for an ethics review.

**Experimental Designs Or Analyses:**

The baselines are too weak.

**Methods And Evaluation Criteria:**

For baselines, the only use of LD Score Regression is too weak. The paper mentioned that a bunch of approximation approaches, they should be considered in verifying the efficiency of the WASP.

**Other Comments Or Suggestions:**

No other comments. The paper would benefit from clearer organization.

**Other Strengths And Weaknesses:**

Weakness:
There is no clear framework of the proposed WASP.
No theoretical guarantee for the convergence.
Only one dataset and one baseline.

**Questions For Authors:**

For this paper, I am still not clear how to involve the informed priors to improve the accuracy in prediction?

**Relation To Broader Scientific Literature:**

The

**Theoretical Claims:**

There are no theoretical guarantees.

---

> ### Author Rebuttal · Authors · 2025-04-01
>
> Thank you for your review. In our paper, we train the first large scale models on recently released huge genomics and functional datasets, enabled by modern linear algebra techniques. Below we address your points on additional baselines and downstream uses of our model.
>
> **On weak baselines**. LD score regression is the de facto framework that is currently used for modeling genetic variability due to its tractable computational demands. Its simplest version – where f is assumed to be constant --  was indeed published ~10 years ago as we cited, but there has been continued research on it ([example](https://doi.org/10.1101/2025.03.07.25323578)).
>
> Indeed, no prior work has been able to train at the scale that we have! Even in the original LD score regression paper, they considered a simple model that could be fit with a closed form formula. Therefore, we devised a gradient descent method to train a large model using LD score regression. As we show, this unfortunately does not work well, necessitating exact likelihood inference and therefore our development of WASP.
>
> The fact that there is no better alternative method to LD score regression highlights the importance and impact of our contribution.
>
> Next, we clarify that the methods in lines 179 and 212 are not competitors to WASP. The methods in line 179 take pre-trained functionally-informed priors and use them to improve genetics tasks – they therefore complement our development of WASP, which is a method for pre-training functionally-informed priors.
> Moreover, we build methods for fast linear algebra operations on large LD matrices. The methods on line 212 include other methods for speeding up linear algebra problems in genetics; genetics is a broad field and these techniques do not solve the problems involved in functionally-informed priors. In particular, they approximate $R$ without necessarily describing how to invert the matrix in the WASP likelihood, $A_\phi^{(i)}$. As an example, we cite Berisa, T. and Pickrell, J. K 2016 who discovered the block-diagonal approximation of $R$ to learn about biophysical processes in the cell. We adapted their discovery to devise our mini-batching strategy for WASP.
>
> **On one dataset.** UKBiobank is the only dataset that satisfies two considerations: 1) the summary association and LD data is public while other databases such as China Kadoorie Biobank are not, 2) UKBiobank is a huge dataset of hundreds of thousands of individuals that we expect to benefit from a larger model. Put differently, UKBiobank is a foundational dataset similar to OpenWebText. In contrast to vision models, where there are a myriad of datasets (CIFAR, SVHN, ImageNet, etc), the genetics datasets are naturally much larger and most publications focus on fitting to the largest amount of data available.
>
> **On informed priors and disease prediction accuracy.** The improved posterior that WASP provides can be used for a myriad of downstream tasks such as disease prediction (simply take a MAP estimate of the effect of each variant) but also for causal variant identification, or to interpret the importance of the diverse functional annotations that we used (Phylo, Encode, Fantom). Indeed the goal of WASP is to solve the machine learning problem of fitting the data in Eqn2 as accurately as possible and, in our previous work section, we highlight some of the many efforts that use the functionally informed priors (that result from solving the prediction problem) for downstream applications that we just mentioned.
>
> **On lack of framework and theoretical convergence.** Succinctly, WASP proposes a loss function Eqn 2 and a computationally efficient way to train the loss in section 4. The approximation that we used for mini-batching has been well studied as cited [Berisa & Pickrell, 2016 & Salehi Nowbandegani et al. 2023) and all the convergence of our linear algebra techniques is well-understood (Saad 2011 or Hobgen, 2013). Additionally, WASP is a consistent estimator as explained on the first point [here](https://openreview.net/forum?id=oOtdWiLb1e&noteId=Uw5dMoGCtr). Thus WASP is a principled method for computing the likelihood in Eqn 2. If your theoretical convergence concern is about our use of neural networks then that is an empirical question that we address with the performance on held-out data.

---

### Official Review · Reviewer_zKNT · 2025-03-13

**Overall Recommendation:** 3

**Summary:**

The paper addresses the challenge of predicting how genetic variants affect phenotypes from large datasets. LD score regression, make simplifying assumptions to avoid computationally expensive linear algebra across genomic metrics. Their method leverages preconditioned linear algebra and GPU acceleration for improved results.

**Claims And Evidence:**

Yes. This paper is out of my area of expertise, so I may not be the best judge.

**Essential References Not Discussed:**

na

**Experimental Designs Or Analyses:**

The authors perform experiments on semi-synthetic data and real data. They are well motivated, however variances are missing. Some of the results can be very close, running multiple seeds might allow for better comparision

**Methods And Evaluation Criteria:**

Performance is evaluated using semi-synthetic data (controlled experiments) and real-world GWAS data (UK Biobank).

The comparison is mainly with LDSR, there could be other baselines potentially, like vae? Is this part of the literature?

**Other Comments Or Suggestions:**

- The different metrics such as BMI, Athama, etc do not have an introduction in the paper. Some context might be more helpful to better understand the significance of the numbers. To a layperson (aka me, I thought BMI is supposed to be between 18-25 but the numbers are much higher here)

**Other Strengths And Weaknesses:**

S1. The authors aim to tackle a realistic problem statement and offer solution in O(n^2) as compare to slower O(n^3).

S2. Instead of using precomputed LD scores, WASP directly optimizes the marginal likelihood, which theoretically leads to better parameter estimates.

W1. They use chromosome-level mini-batching, but it's unclear how it performs on whole-genome datasets.

W2. The paper is missing uncertainty quantification.

**Questions For Authors:**

1. The paper is motivated for genomic studies, however they mainly focuses on chromosome-level analysis rather than whole-genome scalability. Is this a good psuedotask?

2. Are non gaussian priors explored?

**Relation To Broader Scientific Literature:**

The contributions of this paper fit into the broader landscape of genome-wide association studies (GWAS) and Bayesian modeling.

**Theoretical Claims:**

The maths makes sense.

---

> ### Author Rebuttal · Authors · 2025-04-01
>
> Thank you for your review! In our paper, we train the first large scale models on recently released huge genomics and functional datasets. Below we address your points on additional baselines and extensions of our model.
>
> **On additional baselines.** Note there are no methods that have been able to train models at the scale we do. Even in the original LD score regression paper, they considered a simple model that could be fit with a closed form formula. Therefore we devised a gradient descent method to train a large model using LD score regression. As we show, this unfortunately does not work well, necessitating exact likelihood inference and therefore our development of WASP.
>
> Indeed there are variational inference methods for fitting these models among our citations. However, these models were at best used to fit a handful of hyperparameters, sometimes using grid search; there is no straightforward way to adapt them to scalably fit a large scale model using gradient descent. In particular, this is because one must update the posteriors over the effect sizes over the whole genome before taking a step to update the hyperparameters.
>
> **On whole-genome datasets.** Our mini-batching on the SNP level, similar to SGD, is used to accelerate training. However, our resulting model applies to the whole genome. Note we showed that LDSR makes an extreme version of the WASP mini-batching assumption (Sec. 4.1) and is still used for interpreting the whole genome!
>
> What is the connection between models that predict disease from a whole genome and WASP, which is trained on subsets of chromosomes? We begin with a whole-genome generative process that explains the traits of each individual $y$, (Eqn. 1) and we multiply by $X$ to get a generative model of associations on chromosomes $\hat\beta$ (Eqn. 2). The likelihood of these two models is only a constant away from each other, so doing well on one is equivalent to doing well on the other. In particular, we explain below how to interpret our metrics in Tables 1 and 2 as our ability to do whole-genome prediction using the held-out chromosomes.
>
> **On more seeds**. We didn’t provide multiple runs as fitting each model is computationally expensive. Here, we ran two more seeds for Enformer trained on BMI via LDSR and WASP. We saw a standard deviation in log likelihood of 14 for LDSR and 8 for WASP, which is much smaller than the differences in the performance of different methods: 96. Nevertheless, we will report error bars for all reported numbers in future drafts.
>
> **On non Gaussian priors**. Although the Gaussianity assumption is by far the most popular in the population genetic literature, in our citations we include references to methods with sparsity-inducing non-Gaussian priors. Applying modern linear algebra techniques like WASP could help us train large models with these priors in future work!
>
> **On metric interpretation**. We take, for each variant $m$ in held-out chromosomes 21 and 22, its observed association with BMI, height, and asthma $\hat\beta_m$; then we try to explain this data using our Gaussian model in Eqn 2. The metric is the difference likelihood of these associations under a trained prior $f$ vs a model that assumes there’s no genetic effect on these traits ($f_m=0$ for all $m$). It can therefore be interpreted as how well our trained models explain the associations on held-out chromosomes.
>
> Another interpretation is the likelihood of how well we could predict traits in the UKBiobank cohort $y$ if we were to only use held-out chromosomes 21 and 22 for our predictions. This is because the models in Eqns 1 and 2 are equivalent. In practice, to do whole-genome estimation, we would train a different model holding out each chromosome and combine their predictions on each held-out chromosome; the likelihood of such a “whole-genome” model would be the sums of likelihoods like those in tables 1 and 2.
>
> This is why the numbers we show are not in the regular BMI range. We will elaborate on this explanation in the paper.

---

### Official Review · Reviewer_fAUH · 2025-03-15

**Overall Recommendation:** 3

**Summary:**

This paper introduces WASP, a method leveraging accelerated linear algebra to train flexible neural network models for predicting genetic variant effects from functional annotations. By employing banded LD matrix approximations and a structured preconditioner, WASP efficiently handles large-scale genomic data while avoiding costly matrix inversions. Experimental results demonstrate superior performance over LD score regression in predicting phenotypic associations and recovering causal variants. The approach enables training larger models with richer genomic features, advancing functionally informed priors for polygenic trait analysis.

## update after rebuttal.
      The authors have provided additional results and explanations. I have raised the score accordingly.

**Claims And Evidence:**

The authors use semi-synthetic data to validate WASP, but the data is generated by an Enformer-based model, which biases the evaluation in favor of neural networks and does not prove real-world generations. A stronger validation would involve testing on data generated by different models or real causal variants.

**Essential References Not Discussed:**

I did not identify any major omissions in the reference.

**Experimental Designs Or Analyses:**

The paper lacks a comprehensive model comparison that focuses on both computational complexity and performance. In Figure 2, the authors compare the calculation complexity on WASP and other methods (no preconditioner, Nystrom and Cholesky). However, in Table 1, the authors compare the performance on WASP and LDSR. To provide a more balanced comparison, the authors may consider adding a computational complexity evaluation on LDSR. This would allow them to demonstrate that WASP outperforms LDSR not only in terms of performance but also in computational efficiency, presenting a more holistic view of its advantages across all relevant aspects.

**Methods And Evaluation Criteria:**

Yes.

**Other Comments Or Suggestions:**

The writing of the paper still needs to be improved. For example, there is a lack of punctuation: “however there are orders of…” should be “however, there are orders of…”; “Unfortunately this is numerically” should be “Unfortunately, this is numerically”. Typo errors: “we the use well established” should be “we then use well established”.

The tables in the paper are labeled as “Table 1” and “Table 2”, but in the text they are referred as “Table 6.3”, which might lead to confusion.

**Other Strengths And Weaknesses:**

This paper effectively integrates neural network training with modern fast linear algebra techniques, providing a computationally efficient approach to GWAS data analysis. This combination is a promising direction for accelerating model training and improving scalability in genomic studies.

However, the experimental results and analysis can be more diversified and detailed. One potential limitation is the lack of discussion on the impact of sliding window algorithm and the window size on the final result. The method assumes that only variants inside the window are considered, but this choice could slightly influence model accuracy and interpretation. A deeper analysis of how window size - especially in regions spanning different chromosomes or chromosomal structures - affects performance would strengthen the paper’s claim.

**Questions For Authors:**

How exactly is WASP incorporated into the model training process?
The paper provides mathematical formulas showing how WASP works. However, it does not clearly specify when, where and how WASP is applied in the training pipeline. Is it used during forward propagation, back propagation, loss computation, or another stage? A detailed explanation or a pipeline diagram would greatly improve clarity.

**Relation To Broader Scientific Literature:**

By adapting iterative algorithms and structured preconditioners from Gaussian process literature, WASP overcomes the bottleneck of LD matrix inversion to accelerate linear algebra operations.

**Theoretical Claims:**

I have reviewed the formulas and theoretical claims presented in the paper. Based on my examination, I did not find any major flaws or inconsistencies in the mathematical derivations.

---

> ### Author Rebuttal · Authors · 2025-04-01
>
> Thank you for your detailed and insightful review. We ran your suggested experiments which have significantly strengthened our paper.
>
> **On semi-synthetic simulations.** You raised a fair point that having a randomly initialized Enformer as a ground-truth model might benefit WASP. Thus, we re-ran all the semi-synthetic simulations but now set a biologically-informed model of inheritance as the ground-truth function. That is, we considered a threshold-based model that whenever the sum of a track in a window around a variant is above a threshold, this track adds a multiplicative increase to the effect size: $$f_{\theta, m}=\sum_{d}\mathbb{1}\left(\left(\sum_{w}C_{\mathrm{track}, m, d, w} \right)> \mathrm{thresh}_d\right)\times\mathrm{enrich}_d.$$
> We chose the parameters such that $f$ is sparse (mostly only 4 tracks pass the value $\mathrm{thresh}_d$) and multi-modal (with randomly selected multipliers $\mathrm{enrich}_d$). Indeed, given the flexibility of the Enformer NN, WASP approximates the correct $f$ much better than other alternatives used in practice and that we compare to as seen in this [figure](https://drive.google.com/file/d/1mf-xaMQnK3znZLDeFI1nIeTT5FDkF4rL/view?usp=sharing). This experiment better highlights the impact of having a flexible neural network model.
>
> **On computational complexity.** WASP is roughly 36% more computationally expensive than LD score regression due to the likelihood computation. However, the discrepancy between WASP and LD score regression is not that large as the cost of the likelihood is not as expensive as the forward and backward pass of the NN that both LD and WASP have to do. Moreover, increasing the model size would decrease the gap. We will discuss this runtime consideration in the paper. Also, to clarify, the goal of section 4 and, in particular, Figure 2, is to discuss alternatives techniques to most efficiently compute the likelihood for WASP.
>
> **On the effect of window size**. We make two distance-based approximations: (1) we break chromosomes into windows of size 1,000,000 and treat observed associations in each window as independent of other associations, and (2) there is no LD for variants further than 1,000,000 positions away from each other.
>
> Assumption (1) is mostly of no concern as WASP would still be a consistent statistical estimator, that is, given enough data the estimator will converge to the correct optimum. Intuitively, this is the case as we train our model using parts of the evidence and treating all the other variables as missing (a proof can be found on the LD score regression paper). Moreover, we do not expect our method to be very sensitive to this parameter, and shrinking the window size should interpolate between the behavior of WASP and LDSR since LDSR is WASP with a window size of 1 (section 4.1).
>
> According to prior work we took a very conservative approach to assumption (2) by having no LD variants further than 1M positions away. However, to probe your question we trained a model that incorporates a stronger assumption where variants that are more than 100,000 positions from each window have no LD with variants inside the window. We noted that there is no significant drop in performance on the test set: the likelihood only dropped by 14, while training with LDSR instead of WASP drops the likelihood by 96, and using a linear rather than an Enformer architecture drops the likelihood by 51. Finally, we do find this analysis to be interesting and we’ll expand it on the paper with other window changes.
>
> **WASP on training process.** WASP is involved in the forward and backward pass of the loss computation. Given a mini-batch of $\beta$ and tracks (where the mini-batching could arguably be considered part of WASP), WASP computes the log determinant and quadratic terms of the loss using fast linear algebra methods. For the backward pass WASP uses a stochastic estimator of the loss. Below we’ve added pseudocode to illustrate these two steps.
>
> // Pseudocode for WASP fwd
>
> 1: $\log(f)$ = NN(geno, anno)  // compute NN output for mini-batch
>
> 2: $A = R F R + \sigma^2 R + \epsilon I$ // form matrix section 4.1
>
> 3: $ P $   // compute preconditioner from section 4.4
>
> 4: $\log(|A|) = SLQ(A, P)$ // compute see section 4.3
>
> 5: $A^{-1} \beta = CG(A, P, \beta)$ // compute see section 4.3
>
> 6: $\ell = log(|A|) + \beta^T A^{-1} \beta$  // compute log-likelihood
>
> 7: return $\ell$
>
> // Pseudocode for WASP bwd
>
> 1: $u_i \sim N(0, I)$   // sample M probes
>
> 2: $ \frac{1}{M} \sum_i \text{VJP}(u_i, A, A^{-1} u_i ) = \frac{1}{M} \sum_i u_i^T \nabla A (A^{-1} u_i) \approx \nabla \log(A) $ // section 4.3
>
> 3: $ \text{VJP}(-\beta^{T}A^{-1}, A, A^{-1}\beta ) = - (A^{-1} \beta)^T \nabla A (A^{-1}\beta) = \nabla \beta^{T} A^{-1} \beta$ // section 4.3
>
> 4: return  $\frac{1}{M} \sum_i u_i^T \nabla A (A^{-1} u_i) - (A^{-1} \beta)^T \nabla A (A^{-1}\beta) $ // gradient of $\ell$

---

### Official Review · Reviewer_j54Y · 2025-03-15

**Overall Recommendation:** 4

**Summary:**

The paper introduces WASP, a method for training large-scale neural network models to predict the effects of genetic variants from functional annotations. The paper begins by introducing linear models for variant effects, the need for good priors, and how they can be fit using anonymous summary statistics. This introduces the challenge of LD matrix $R$ inversion. The paper then introduces a batching approach and a preconditioner for computing the likelihood, overcoming the coputational challenge.

An Enformer model for $f_{\theta}$ is then trained on large scale functional genomic data from ENCODE, FANTOM, PhyloP and ESM-2. The resulting WASP method is validated using UKBB data against altnerative approaches, demonstrating improved performance.

## update after rebuttal
The authors provided additional experiments and clarficiations. My score is unchanged at 4 as this is a good paper that should be accepted.

**Claims And Evidence:**

The idea for batching turns out to be quite straightforward: chunk the genome into blocks, by assuming block-diagonal LD structure. Then again assume locality, with distant variants not correlating.

Both assumptions are presented as-is with relevant citations of prior work. I still wonder whether there is any sensitivity to the distances used here?

**Essential References Not Discussed:**

N/A

**Experimental Designs Or Analyses:**

- How was the size of the Enformer network optimized? Is this a critical parameter - the experiment shows that making it smaller reduces performance, but what happens in the other direction - is there a risk of overfitting, and if so, how would we detect it.

**Methods And Evaluation Criteria:**

Benchmarking approach is adequate, with realistic UKBB data used. Empirical evidence for the used assumptions would be great still.

**Other Comments Or Suggestions:**

N/A

**Other Strengths And Weaknesses:**

- The paper is very well written and introduces all required concepts to clearly motivate the need for WASP. The method itself is elegant, breaking down a numerically intractable problem into chuks that are amenable to minibatch neural network training approaches.

**Questions For Authors:**

- It is unclear to me if there is a contribution in 4.3. Is my understanding correct that this in essence a rationale of what prior work to leverage (without introducing new concepts)?

**Relation To Broader Scientific Literature:**

The paper introduces related work and makes a clear case for what limitations WASP seeks to overcome.

**Theoretical Claims:**

N/A

---

> ### Author Rebuttal · Authors · 2025-04-01
>
> Thank you for your support and thoughtful review! Below we discuss new experiments that address the sensitivity to the LD window distance cutoff, larger model sizes and clarify the purpose of Section 4.3.
>
> **On distance sensitivity and assumptions.** We make two distance-based approximations: (1) we break chromosomes into windows of size 1,000,000 and treat observed associations in each window as independent of other associations, and (2) there is no LD for variants further than 1,000,000 positions away from each other.
>
> Assumption (1) is mostly of no concern as WASP would still be a consistent statistical estimator, that is, given enough data the estimator will converge to the correct optimum. Intuitively, this is the case as we train our model using parts of the evidence and treating all the other variables as missing (a proof can be found on the LD score regression paper). Moreover, we do not expect our method to be very sensitive to this parameter, and shrinking the window size should interpolate between the behavior of WASP and LDSR since LDSR is WASP with a window size of 1 (section 4.1).
>
> According to prior work we took a very conservative approach to assumption (2) by having no LD variants further than 1M positions away. However, to probe your question we trained a model that incorporates a stronger assumption where variants that are more than 100,000 positions from each window have no LD with variants inside the window. We noted that there is no significant drop in performance on the test set: the likelihood only dropped by 14, while training with LDSR instead of WASP drops the likelihood by 96, and using a linear rather than an Enformer architecture drops the likelihood by 51. Finally, we do find this analysis to be interesting and we’ll expand it on the paper with other window changes.
>
> **On Enformer size.** We picked the size of a network that we could fit with reasonable compute. In principle, training a large-enough model for long enough should result in overfitting. Based on your question and our computational resources, we trained a larger Enformer model on the BMI data (we increased the number of attention layers from 2 to 8) and noted slight improvements on the test likelihood (increased by 11) suggesting that we are not yet overfitting. Indeed it appears that there is more potential in leveraging the huge data collected in large genetics efforts with even larger models. Measuring the scaling laws of these models could therefore be an interesting direction for future work.
>
> **On purpose and contributions in Section 4.3.** Indeed Section 4.3 is a review of iterative methods used for optimizing systems of large matrices. In particular, it introduces the identities for the gradients of the matrix inverse and log determinant; these formulas have been previously introduced in [1,2,3]. Although the construction of the preconditioner in section 4.4 is ours. We’ll clarify our contributions and prior work in section 4.3.
>
> [1] Gardner et al. 2018. GPyTorch: Blackbox Matrix-Matrix Gaussian Process Inference with GPU Acceleration.
>
> [2] Saad 2003. Iterative Methods for Sparse Linear Systems.
>
> [3] Saad 2011. Numerical Methods for Large Eigenvalue Problems.

---

### Decision · Program_Chairs · 2025-05-01

**Decision:**

Accept (poster)

**Comment:**

The majority of the reviewers assigned a weakly positive score to this paper. They all seem to agree that the paper is clearly motivated and well-written, that a relevant problem is studied and that the solution provided in the form of a combination of neural networks with fast linear algebra techniques is interesting and technically sound. There were also several more negative comments, such as the some experimental design choices (effect of sliding window, use of chromosome-level mini-batching etc.), missing convergence guarantees and missing uncertainty quantification. Some of these concerns, however, could be addressed in the rebuttal (at least to some extent). In summary, I think that although this paper is still in the "borderline region", to me the positive aspects regarding relevance and technical soundness slightly dominate to points of criticism.